

# Potential relationship of the gut microbiome with testosterone level in men: a systematic review

Cennikon Pakpahan[1], Geraldo Laurus[1], Markus Christian Hartanto[1], Rajender Singh[2], Ankur Saharan[3,4], Darmadi Darmadi[5], Andri Rezano[6] and Gito Wasian[7]

[1] Andrology Study Program, Department of Biomedical Sciences, Faculty of Medicine, Universitas Airlangga, Surabaya, East Java, Indonesia
[2] Division of Endocrinology, Central Drug Research Institute India, Lucknow, Uttar Pradesh, India
[3] Faculty of Medicine, Amity University India, Noida, Uttar Pradesh, India
[4] Department of Human Genetic, McGill University, Montreal, Quebec, Canada
[5] Department of Internal Medicine, Faculty of Medicine, Universitas Sumatera Utara, Medan, North Sumatera, Indonesia
[6] Department of Biomedical Sciences, Faculty of Medicine, Universitas Padjadjaran, Sumedang, West Java, Indonesia
[7] Department of Medical Biology, Faculty of Medicine, Universitas Indonesia, Jakarta, Indonesia

Corresponding author
Cennikon Pakpahan,
cennikon.pakpahan@fk.unair.ac.id

## ABSTRACT

The gut microbiome influences the metabolism and health of several organs beyond the gut and has recently gained considerable attention. The gut plays a central role in food digestion, absorption, nutrition, and general health. Hence, the gut microbiome has become a research subject for almost all health conditions. Imbalance or dysbiosis in the gut microbiome can cause different health issues in men, including reproductive and sexual health issues associated with testosterone levels. Several studies have investigated the relationship between the gut microbiome and testosterone levels. In this systematic review, we aimed to examine the relationship between the gut microbiome and testosterone levels in men. Literature searches were conducted by scanning PubMed, ProQuest, EBSCO, Taylor & Francis Online, Wiley Online, Springer Link, Web of Science, Google Scholar, and Science Direct databases for relevant keywords following the preferred reporting items for the systematic review guidelines. This review included cross-sectional, case-control, retrospective, and prospective cohort studies. Quality assessment was conducted using the Newcastle-Ottawa Scale. We found a significant positive correlation between the gut microbiome and testosterone levels in men. Several microbes play substantial roles in testosterone production. Mechanisms have been proposed as factors that contribute to testosterone levels, namely the hypothalamus-pituitary-gonad axis modulation, androgen metabolism, and intestinal homeostasis, by balancing the bone morphogenic protein (BMP) and the Wnt diverse microbiome. Ruminococcus showed a stronger correlation with testosterone levels than other microorganisms. The gut microbiome has complex correlations with testosterone metabolism. However, the microbiome with the most significant influence on testosterone levels cannot be easily identified and requires further research.

## INTRODUCTION

A microbiome is a collection of all the taxa that constitute the microbial community in a particular organ or system. Bacteria constitute approximately 99% of all microbiomes in the human body, whereas other microorganisms, such as viruses, archaea, protozoa, and fungi, constitute the remaining 1% (*Marchesi et al., 2016*). Several studies have demonstrated the influence of the microbiome in predicting and improving the clinical outcomes of acne vulgaris (*Ruchiatan et al., 2023*), atopic dermatitis (*Menul Ayu Umborowati et al., 2022*), pemphigus vulgaris (*Satriyo & Vidyani, 2024*), neurodevelopment delay (*Fadlyana et al., 2022*), tuberculosis (*Wiqoyah et al., 2021*), and sexual dysfunction (*Darmadi et al., 2024*). The human gut microbiome has the highest density and number of microorganisms, with the small intestine and colon differing in their relative abundances (greater density at the distal end than at the proximal end) (*Marchesi et al., 2016*; *Schroeder & Bäckhed, 2016*). Trillions of microbes inhabit the gut of healthy individuals, with different parts of the gut harboring different microorganisms. The stomach and duodenum contain approximately $10^1$–$10^2$ CFU/mL of the gut microbiome, the jejunum and ileum contain $10^4$–$10^8$ CFU/mL of gut microbes, and the colon with $10^{10}$–$10^{12}$ CFU/mL is the most abundant in gut microbes (*Cresci & Bawden, 2015*). The majority of the gut microbiome comprises only five phyla (*Bacteroidetes, Firmicutes, Actinobacteria, Proteobacteria,* and *Verrucomicrobia*). However, the numbers and species diversity varies considerably. Most of the gut microbiome is comprised of anaerobic bacteria with a mass of 1.5–2 kg (*Schroeder & Bäckhed, 2016*). The remaining 1% of the microorganisms in the human body are found in the respiratory, digestive, integumentary, and reproductive systems (*Kim, 2022*; *Koedooder et al., 2019*).

The gut microbiome is established early in life and remains relatively constant throughout adulthood. It is influenced by several factors such as genotype, body mass index (BMI), lifestyle, physical activity, and dietary and cultural habits (*Koliada et al., 2021*). Metagenomic research has demonstrated that ethnicity, age, stress, psychological factors, antibiotics, health of the pregnant mother, and delivery method influence the gut microbiome. The host and the microbiome communicate bidirectionally and affect each other's functions (*Kim, 2022*). The gut microbiome influences health and disease physiology by affecting the metabolic functions and immune system, protecting against pathogens, and directly or indirectly contributing to various physiological processes (*Shreiner, Kao & Young, 2015*). It also communicates with the brain through the gut-brain axis (*Ranuh et al., 2019*). The gut microbiome is involved in the development, maturation, modulation, and stimulation of the host (*Vemuri et al., 2019*). Furthermore, the gut microbiota regulates metabolism by producing vitamins and short-chain fatty acids (SCFAs). The gut microbiome interacts with the intestine locally; it also interacts with and affects distant organs such as the adipose tissues, liver, pancreas, cardiovascular system, brain, lungs, and reproductive system. Each of these organs forms a network with the intestines, and the gut microbiome influences almost every organ of the human body (*Li et al., 2022*).

The interaction between gut microbiome and reproductive hormones was first introduced by *Flak, Neves & Blumberg (2013)* under the term "microgenderome". In the testicles, Leydig cells produce testosterone, the main sex steroid, at a secretion rate of 7 mg/day (*Hohl, 2023*). The adrenal gland also produces approximately 5% of total testosterone. Testosterone can bind to sex hormone-binding globulin (SHBG) and albumin or circulate freely in plasma. The fundamental mechanism that influences testosterone production is the hypothalamic-pituitary-testicular axis. The gonadotropin-releasing hormone secreted by the hypothalamus stimulates the pituitary gland to release luteinizing hormone, which then stimulates the testes to produce testosterone (*Hohl, 2023*). Testosterone is essential for masculinization, normal sexual function, and spermatogenesis (*Brinkmann, 2011*; *MacLeod et al., 2010*). It targets the brain, skin, muscle, bone, hair follicles, and hematopoietic system. This vast biological effect demonstrates that testosterone is important for the overall health of men, including mood, energy levels, and cognitive function (*Li et al., 2022*). Studies have suggested a possible relationship between the gut microbiome and androgen levels in men. Androgens can profoundly alter gut flora *via* a complex pathway (*Harada et al., 2016*). *Yan et al. (2024)* reported a causal effect of SHBG on gut microbiota. Higher SHBG levels in men were associated with *Dorea* and *Clostridiales.* Genome-wide association studies have shown that *Alphaproteobacteria* are associated with high levels of SHBG, a carrier of sex hormones (*Yan et al., 2024*). The gut microbiome may also influence testosterone production and metabolism through various mechanisms (*Colldén et al., 2019*). Emerging evidence suggests that the gut microbiome influences testosterone production through various mechanisms. One proposed pathway involves the modulation of the hypothalamic-pituitary-gonadal (HPG) axis, where the gut microbiota may affect the release of gonadotropin-releasing hormone (GnRH) and, consequently, luteinizing hormone (LH), which stimulates testosterone synthesis in the testes (*Chen et al., 2024*). Additionally, certain gut microbes possess steroid-processing enzymes that can directly affect androgen metabolism, contributing to testosterone levels in circulation (*Tang et al., 2024*; *Zou et al., 2024*). In contrast, testosterone and other hormones influence the gut microbiome. Diet, lifestyle, toxins, and drugs, among other factors, can influence the gut microbiome (*Li et al., 2022*). Presently, no conclusive evidence is available supporting a direct link between the microbiome and testosterone levels, and the microgenome theory remains unproven. The gut microbiome may also affect sexual and reproductive problems in men, including infertility.

Notably, researchers have conducted a comprehensive analysis to evaluate the association between the gut microbiome and sex hormone levels in women (*D'Afflitto et al., 2022*); however, a similar analysis has not been conducted for men. In this systematic study, we aimed to determine the significance of microgenderomes and the probable relationship between the gut microbiome and testosterone levels in men.

## MATERIALS AND METHODS

### Search strategy and selection

We conducted a systematic review following the Preferred Reporting Items for Systematic Review and Meta-Analysis (PRISMA) 2020 guidelines (*Page et al., 2021*) and registered

this review in the PROSPERO database (CRD42022350064). In this systematic review, we aimed to assess the literature addressing the association between male testosterone levels and the gut microbiome without setting any filters on the publication date. We searched PubMed, ProQuest, EBSCO, Taylor and Francis Online, Wiley Online, Springer Link, Web of Science, ScienceDirect, and Google Scholar databases using the MeSH terms ''gut microbiome'' or ''microflora'' and ''testosterone'' or ''sex steroid'' and ''male.''

## Inclusion and exclusion criteria

This review included observational studies (cross-sectional, case-control, retrospective, and prospective cohorts). We included studies with male subjects or any study on the microbiome that included male subjects, those involving testosterone or sex steroid examinations, and those published in English. Our main finding was the association between the gut microbiome and testosterone levels, which was evident with and without statistical analysis. Duplicate, non-human, irrelevant, and review articles were excluded.

## Search strategy, study selection, and data extraction

Three authors (CP, GL, and MCH) performed keyword searches of the specified databases. All studies that matched the keyword findings were exported to the Mendeley reference manager. Duplicate screening was performed using Mendeley, followed by screening of titles and types of articles; two authors independently evaluated the articles using abstracts. In case of disagreement during screening/data extraction, the authors discussed it internally until an agreement was reached. For example, the design of the study included and the results. The authors also asked experts who were not included in the group for consideration. The PRISMA flowchart shows the study exclusion process and related reasons (Fig. 1).

Data were extracted from all eligible studies (CP, GL, and MCH). The first author's name, publication year, research type, inclusion/exclusion criteria, sample size, dominant microbiome type, and study outcomes were recorded in an Excel spreadsheet. This information was obtained from the individual studies. The authors were contacted to obtain the missing information, if any. Statistical significance of the results was determined using appropriate measures of association, such as relative risk (RR), odds ratio (OR), or correlation coefficient (r), along with corresponding $p$-values. All authors collectively examined all the findings including those that were challenging to interpret. We performed qualitative analyses and comparisons without conducting quantitative analyses because statistical analyses and presentations differed across the studies.

## Quality assessment

We evaluated the quality of the included studies using the Newcastle–Ottawa scale (*Ottawa Hospital Research Institute, 2024*). Studies with a score ≥ of seven were included in this review.

## Ethical approval

No ethical approval was deemed necessary given the nature of the review.

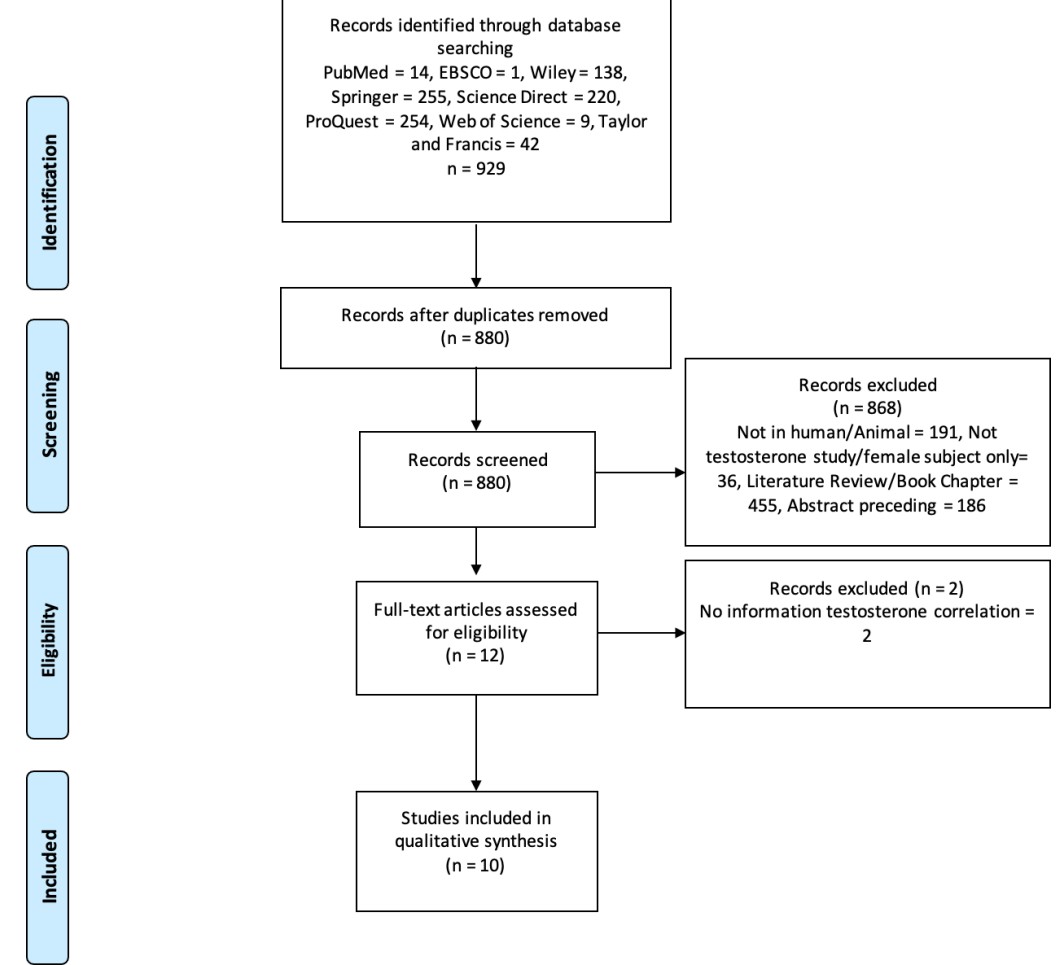

**Figure 1  PRISMA flowchart.** The process of excluding research studies and the reasons for exclusion.

## RESULTS

### Study findings and characteristics

We examined 929 studies from eight databases. After screening, ten studies were deemed appropriate for qualitative analysis. Figure 1 shows the selection procedure for eligible studies and the reasons for exclusion. The ten eligible studies were observational and analytical studies published between 2019 and 2024. According to the Newcastle–Ottawa scale for the qualitative assessment of studies, four were of high quality (score 9) and six were of medium quality (score 8). Therefore, the results or interpretation of each included studies in this review were adequate. These studies were conducted on seven populations: Spanish, Japanese, Ukrainian, Australian, American, Chinese, and Korean. The total sample size in these studies was 35.904 males, whereas one study used an *in vitro* procedure. Some of these studies also collected data from women. However, our analysis focused only on male participants. Therefore, we used data from men for the qualitative analysis, as the review focused on male populations. Table 1 presents the details and characteristics of

the included studies. Our assessment of the quality of all studies included using the New Ottawa Scale $\geq 8$, indicating that the included studies were of good quality for review.

## Outcome

We examined the relationship between the gut microbiome and testosterone levels in men (Table 2). However, a quantitative analysis was not performed in this review because each study employed different parameters. These six studies differed in their reporting of the dominant microbiome types in their subjects. Furthermore, some studies provided only phyla, rather than microbiome species (*Jie et al., 2021*; *Matsushita et al., 2022b*; *Shin et al., 2019*). A qualitative analysis of these studies suggests that the gut microbiome has a significant impact on testosterone levels in men. *Wilmanski et al. (2019)* discovered that the gut microbiome positively correlated with testosterone's metabolic outcome ($5\alpha$-androstane-$3\beta$-$17\alpha$-diol disulfate), and *Bacteroides* were anti-correlated with this metabolite. Similarly, *Shin et al. (2019)* reported that *Acinetobacter* ($r = 0.3782$, $p = 0.0359$), *Dorea* ($r = 0.3976$, $p = 0.0268$), *Megammonas* ($r = 0.4161$, $p = 0.0199$), and *Ruminococcus* ($r = 0.4589$, $p = 0.0094$) correlated with testosterone levels in men. *Jie et al. (2021)* found that the gut microbiome had the greatest predictive power for metabolites and plasma hormones, including testosterone. *Matsushita et al. (2022b)* reported an association between the gut microbiome and testosterone levels in a Japanese population and testosterone levels, indicating that the number of *Firmicutes* in the gut correlated with serum testosterone levels ($r = 0.3323$, $p = 0.0141$) (*Matsushita et al., 2022a*). These reports are consistent with the finding that the gut microbiome affects testosterone levels.

In each study, several microbial types, such as the phyla *Bacteroidetes* and *Firmicutes* were consistently prevalent (*Jie et al., 2021*; *Koliada et al., 2021*; *Matsushita et al., 2022b*; *Mayneris-Perxachs et al., 2020*; *Shin et al., 2019*; *Wilmanski et al., 2019*). No other phyla were identified in any of the qualitative investigations. In contrast, *Mayneris-Perxachs et al. (2020)* examined an orthogonal partial least squares model to predict circulating testosterone levels and reported that the gut microbiome could not predict circulating testosterone levels.

The gut microbiome influences testosterone levels, and testosterone controls gut microbiome diversity. This is consistent with the findings of *Matsushita et al. (2022b)*, who reported a more significant number of gut microbes in the high-testosterone group than in the low-testosterone group. *Koliada et al. (2021)* reported similar findings, in which the type and number of specific microbiomes differed between men and women. This is considered valid because testosterone levels differ significantly between men and women (*Koliada et al., 2021*). The prevalence of the gut microbiome may be influenced by comorbidities, obesity, dietary patterns, and lifestyle factors.

All the included studies had different microbiomes and statistical outputs; therefore, we could not continue the quantitative analysis or meta-analysis.

**Table 1 Characteristics of the studies included in the review.**

| Author, year | Type of study | Country of origin | Sample size | Inclusion/exclusion criteria | Quality score of study |
|---|---|---|---|---|---|
| Shin et al. (2019) | Cross-sectional | The Republic of Korea | 57 | Men aged between 25 and 65 years. Participants who frequently consumed pre- or probiotics, antibiotics, lipid-lowering, or glucose-regulating medicines within six months preceding sampling were excluded. In addition, men with active heart disease, cancer, diabetes, inflammatory diseases, or gastrointestinal problems, were disqualified. | 9 |
| Wilmanski et al. (2019) | Cross-sectional | USA | 111 | Unspecified sample criteria, but the study focused on body mass index (BMI) with gut microbiome and metabolome. | 8 |
| Mayneris-Persxachs et al. (2020) | Cross-sectional | Spain | 42 | Inclusion: BMI participants with obesity (BMI 30 kg/m²) and age- and sex-matched participants without obesity. Exclusion criteria: the use of antibiotics, antifungals, antivirals, or treatment with proton-pump inhibitors; severe disorders of eating behavior or major psychiatric antecedents; excessive alcohol consumption (80 g OH/day in men); type 2 diabetes mellitus; chronic inflammatory systemic diseases; acute or chronic infections in the previous month. | 8 |
| Matsushita et al. (2022b) | Cross-sectional | Japan | 54 | Older healthy Japanese male individuals. The men suspected of having prostate cancer underwent prostate biopsies and were eliminated from the study if the results were positive. | 9 |
| Koliada et al. (2021) | Cross-sectional | Ukraine | 786 | Fecal samples from healthy individuals, excluding men with infectious diseases, cancer, cognitive impairments, type 1 diabetes, or poorly controlled type 2 diabetes, consumed prebiotics, probiotics, antibiotics, or immunosuppressants. | 9 |

**Table 1** (*continued*)

| Author, year | Type of study | Country of origin | Sample size | Inclusion/exclusion criteria | Quality score of study |
|---|---|---|---|---|---|
| Jie et al. (2021) | Cohort Prospective | China | 3,400 | No specific inclusion criteria was mentioned. Healthy men with complete data of questionnaire, disease history, marital status, eating, and exercise habits were included. | 9 |
| Yan et al. (2024) | Bidirectional Mendelian randomization design | China | 31,333 | No specific inclusion criteria was mentioned. Healthy men with complete data of sex hormone-binding globulin levels were included. | 8 |
| Ridlon et al. (2013) | *in-vitro* study | Japan | – | This study focused on *in-vitro* study using *C. scindens* ATCC 35704. Prior to RNA isolation, the steroid-17,20-desmolase activity in *C. scindens* ATCC 35704 was confirmed using an HPLC experiment. | 8 |
| Liu et al. (2022) | Observational study | China | 46 | Male patients with Type 2 Diabetes Mellitus. Patients were excluded if they were >60 years; had history with hypogonadotropic hypogonadism, thyroid dysfunction, ramex, testicle injury, epididymis injury, abnormal karyotype, or other diseases that can affect testosterone levels; use of antibiotics or hormone therapy within the last 6 months; patients with diabetic ketoacidosis or diabetic non-ketotic hyperosmolar syndrome. | 8 |
| Tremellen (2016) | Experimental study | Australia | 75 | Patients aged between 18 and 50 years. Exclusion criteria: history of immunosuppressive drug or supplement use (non-steroidal anti-inflammatory drugs, corticosteroids, fish oil), documented primary hypogonadism (Klinefelter's Syndrome, cryptorchidism, or testicular injury), inflammatory or infectious diseases, and any hormonal therapy (aromatase inhibitors, clomiphene citrate, human chorionic gonadotropin, or testosterone). | 8 |

**Table 2 Qualitative analyses of the studies included in the review.**

| Author, year | Most microbiome diversity identified in men | Results (the correlation with testosterone) |
|---|---|---|
| *Shin et al. (2019)* | *Acinetobacter, Dorea, Ruminococcus, Firmicutes, Megamonas, Bacteroidetes, Verrucomicrobia, Synergistetes, Bacteroidetes, Lentisphaerae, Firmicutes, Tenericutes, Fusobacteria, TM7, Proteobacteria, Cyanobac- teria, and Actinobacteria* | • No significant changes in the richness determined by Chao1 were found among the groups of low, medium, and high levels of testosterone<br>• Men with high testosterone levels exhibited significantly greater diversity and evenness in their microbiome compared to men with medium testosterone levels ($p = 0.0462$ for the Shannon index and $p = 0.0408$ for the Simpson's E index)<br>• *Acinetobacter* ($r = 0.3782, p = 0.0359$), *Dorea* ($r = 0.3976, p = 0.0268$), *Megammonas* ($r = 0.4161, p=0.0199$), and *Ruminococcus* ($r = 0.4589, p = 0.0094$) were positively correlated with the level of serum testosterone. |
| *Wilmanski et al. (2019)* | *Firmicutes (Ruminococcaceae Ruminococcus, Ruminococcaceae Oscillospira, Ruminococcaceae unclassified, Clostridiales Unclassified, Lachnospiraceae Roseburia, Ruminococcaceae Faecalibacterium, Lachnospiraceae Lachnospira), Proteobacteria (Alcaligenaceae Sutterella), Bacteroidetes (Bacteroidaceae Bacteroides), etc.* | • A positive correlation was observed between Shannon diversity and 5-androstan-3-17-diol disulfate (testosterone metabolite).<br>• Plasma concentrations of 5α-androstan-3β- 17α–diol disulfate were significantly higher in men compared to those in women.<br>• In addition, the largest positive predictor, 5-androstan-3-17– diol disulfate and the Bacteroides genus were anticorrelated, demonstrating a connection between the dominance of this taxon and the lack of Shannon diversity in our cohort. |
| *Mayneris-Perxachs et al. (2020)* | *Bacteroidetes (Prevotella sp. CAG:520, Prevotella multisaccharivorax, Prevotella sp. AGR2160, Prevotella nanceiensis, Prevotella sp. HMSC073D09, Prevotella bergensis, Prevotella sp. CAG:924, Prevotella ruminicola, Cytophagaceae, Sphingobacteriaceae, Flavobacteriacea) Firmicutes (Clostridium sp. CAG:277), Pseudomonodota (Pasteurellaceae, Idiomarinaceae).* | • *Actinobacteria, Proteobacteria, Firmicutes*, and *Verrucomicrobia* phylum were negatively associated with testosterone levels, while *Prevotellaceae, Cytophagaceae, Fibrobacteriaceae, Sphingobacteriaceae*, and *Idiomarinaceae* were positively associated with testosterone levels.<br>• While adjusting with age and obesity, *Fibrobacteriaceae, Idiomarinaceae*, and *Bacteroidetes* phylum (*Sphingobacteriaceae, Cytophagaceae, Prevotellaceae*, and *Flavobacteriaceae*) had positive associations with testosterone levels.<br>• Predictive circulating testosterone levels using the Orthogonal Partial Least Squares model were lower ($Q^2 Y = 0.04, p = 0.003$) and gut microbiota composition of participants with obesity could not predict the circulating concentrations of these gonadal steroids ($Q^2 Y = -0.12$ and $Q^2 Y = -0.25$).<br>• *Cytophagaceae, Prevotellaceae, Fibrobacteriaceae, Sphingobacteriaceae, Flavobacteriaceae*, and *Idiomarinaceae* consistently positively correlated with testosterone concentrations in a prospective one year study. |

**Table 2** (*continued*)

| Author, year | Most microbiome diversity identified in men | Results (the correlation with testosterone) |
|---|---|---|
| *Matsushita et al. (2022b)* | *Bacteroidetes, Firmicutes, and other small microbiota* | • Testosterone levels do not impact the gut microbiota diversity.<br>• Nine bacterial taxa were significantly more abundant (especially *Firmicutes*) in the gut microbiota in high group testosterone, and six taxa were significantly less abundant in the TT high group ($p < 0.05$, LDA (Linear discriminant analysis) score $\geq |2.0|$)<br>• A correlation is found between the number of *Firmicutes* in the gut and serum testosterone, as determined by Spearman's rank correlation coefficient ($r = 0.3323$, $p = 0.0141$). |
| *Koliada et al. (2021)* | *Firmicutes (Clostridium perfringens), Actinobacteria, Bacteroidetes (Bacteroides thetaiotaomicron), Verrucomicrobiota (Akkermansia muciniphila), Bacillota (Faecalibacterium prausnitzii, Parvimonas micra), Fusobacteriota (Fusobacterium nucleatum), etc.* | • No specific report found on the correlation between those microbiota and sex steroid levels. The study recounts that *Actinobacteria* are the most abundant in men than in women ($p = 0.023$).<br>• This report suggests that gut microbiota plays a key role in maintaining normal testosterone levels. |
| *Jie et al. (2021)* | *Firmicutes (Oscillibacter, Clostridium sp. AT4, Oscillibacter sp. KLE1745, Anaerovoracaceae), Bacteroidetes (Alistipes obesi, Alistipes shahii, Odoribacter splanchnicus), Fusobacterium (Fusobacterium mortiferum)* | • The gut microbiome showed the greatest predictive power for metabolites plasma hormones (testosterone, aldosterone, and random forest cross-validation $R\frac{1}{4}$ 0.36 and 0.20).<br>• While an initial analysis showed a positive association between Fusobacterium mortiferum and testosterone levels, this association was only observed in male participants. However, this association was not statistically significant after adjusting for multiple comparisons |
| *Yan et al. (2024)* | *Coprobacter, Ruminococcus, Barnesiella, Actinobacterium, Eubacterium fissicatena, Alphaproteobacteria, Dorea, Clostridiales* | • *Coprobacter, Ruminococcus, Barnesiella, Actinobacterium, Eubacterium fissicatena* correlated with lower SHBG levels in men, and *Alphaproteobacteria* correlated with higher SHBG levels. |
| *Ridlon et al. (2013)* | *Clostridium scindens* | • The *in-vitro* study showed that steroid product was formed *by C. scindens* ATCC 35704. This micorbiome could convert glucocorticoids to androgens by side-chain cleavage. This process was determined by $11\beta$-hydroxyandrost-4-ene-3,17-dione ($11\beta$-OHA). This study was not specific to the male population, however the significant results provide the possibility of the gut microbiome in steroidogenesis. |

**Table 2** (*continued*)

| Author, year | Most microbiome diversity identified in men | Results (the correlation with testosterone) |
|---|---|---|
| *Liu et al.* (2022) | *Blautia, Lachnospirales, Firmicutes, Lachnoclostridium, Bergeyella, Streptococcus* | • Several microbiomes such as *Blautia* and *Lachnospirales* were higher in males with testosterone deficiency. *Firmicutes* and *Lachnospirales* had negative correlation with testosterone levels. C–reactive protein (CRP) and homeostatic model assessment of insulin resistance (HOMA-IR) improvement did not change the negative correlation of *Lachnospirales* with testosterone level. Meanwhile, at the genus level, *Lachnoclostridium, Blautia,* and *Bergeyella* had a statistically significant negative association with testosterone levels |
| *Tremellen* (2016) | *Escherichia coli* | • Exposure of endotoxin *Escherichia coli* may affect androgen levels in men with obesity. A negative correlation was found between obesity and testosterone, and a positive correlation was found between endotoxin exposure (LBP) and inflammation (C-reactive protein, IL-6). Moreover, a positive correlation was found between IL-6 and endotoxemia (LBP) but a negative correlation between IL-6 and serum testosterone. These findings support the theory that the microbiome has a role in regulating inflammation, energy balance, and metabolism. |

## DISCUSSION

Ten studies reported the relationship between the gut microbiome and testosterone levels. *Wilmanski et al. (2019)* discovered that the gut microbiome positively correlated with a testosterone metabolite (5$\alpha$-androstane-3$\beta$-17$\alpha$-diol disulfate), and *Bacteroides* were anti-correlated with this metabolite. *Shin et al. (2019)* reported that *Acinetobacter* ($r = 0.3782$, $p = 0.0359$), *Dorea* ($r = 0.3976$, $p = 0.0268$), *Megammonas* ($r = 0.4161$, $p = 0.0199$), and *Ruminococcus* ($r = 0.4589$, $p = 0.0094$) positively correlated with testosterone levels in men. *Jie et al. (2021)* found that the gut microbiome had the greatest predictive power for metabolites and plasma hormones. *Matsushita et al. (2022b)* reported an association between the gut microbiome and testosterone levels in a Japanese population, with the specific finding that the number of *Firmicutes* in the gut correlated with testosterone levels ($r = 0.3323$, $p = 0.0141$).

Identifying the specific microbes that predominantly affect testosterone levels could be valuable. Several microbiome types were consistently reported across studies, such as the phylums *Bacteroidetes* and *Firmicutes* (*Jie et al., 2021*; *Koliada et al., 2021*; *Matsushita et al., 2022b*; *Mayneris-Perxachs et al., 2020*; *Shin et al., 2019*; *Wilmanski et al., 2019*). In contrast to the other study findings, *Mayneris-Perxachs et al. (2020)* used an orthogonal partial least squares model to predict circulating testosterone levels in men and found that the gut microbiome could not predict circulating testosterone levels. A quantitative analysis of these study findings would be valuable; however, the data presentation across these studies differed significantly. Furthermore, some studies have reported only phyla rather than specific microbiome species (*Jie et al., 2021*; *Matsushita et al., 2022b*; *Shin et al., 2019*).

Thus, the gut microbiome influences testosterone levels, which may also control gut microbiome diversity. This is consistent with the findings of *Matsushita et al. (2022b)*, who discovered a more significant number of gut microbes in the high-testosterone group than in the low-testosterone group. Androgens maintain intestinal homeostasis by balancing bone morphogenetic proteins and wingless/transforming growth factor-beta signaling in stromal cells (*Yu et al., 2020*). Therefore, the role of androgens in regulating the intestinal barrier and microenvironment may affect and govern the gut microbiome.

Sex hormones, such as estrogen and testosterone, influence gut microbiome diversity (*D'Afflitto et al., 2022*; *Ma & Li, 2019*). High estrogen levels are associated with higher *Bacteroidetes* levels and lower levels of *Firmicutes* and the *Ruminococcaceae* family (*D'Afflitto et al., 2022*). Progesterone boosts *Bacteroides* and *Prevotella intermedius* growth in mice (*He et al., 2021*). The gut microbiome composition of female mice was similar to that of castrated male mice (*He et al., 2021*). Males with high testosterone levels had higher gut microbiome diversity. After puberty, males have relatively lower $\alpha$ gut microbiome diversity than females (*Wang et al., 2019*); however, a Japanese study reported different results (*Matsushita et al., 2022b*). The gut microbiome diversity did not differ significantly between males and females (*Takagi et al., 2019*).

*Shin et al. (2019)* observed that *Acitenobacter, Dorea, Ruminococcus,* and *Megamonas* were prevalent in males with high testosterone levels, whereas *Atopobium* was less common. Of the gut microbiomes, *Ruminoccus* was the genus most significantly associated with

testosterone levels. Other studies conducted under different conditions and locations have revealed significant differences in the gut microbiomes (*Fransen et al., 2017*; *Haro et al., 2016*; *Markle et al., 2013*). *Firmicutes* and *Actinobacteria* were more prevalent in men than in women, whereas *Bacteroidetes* were less prevalent (*Koliada et al., 2021*). *Koliada et al. (2021)* reported similar findings, in which men and women had different types and numbers of specific microbiomes. Sex-specific differences in the gut microbiome may be associated with the differences in hormone metabolism, sexual function, and health profiles between men and women.

*Shin et al. (2019)* reported that *Acinetobacter, Dorea, Megammonas*, and *Ruminococcus* were positively correlated with serum testosterone levels in Korean men. However, *Mayneris-Perxachs et al. (2020)* reported that *Actinobacteria, Proteobacteria, Firmicutes*, and *Verrucomicrobia* were negatively associated with testosterone levels in Spanish men, whereas *Prevotellaceae, Cytophagaceae, Fibrobacteriaceae, Sphingobacteriaceae,* and *Idiomarinaceae* were positively associated with testosterone levels. *Matsushita et al. (2022b)* also reported that *Firmicutes* were positively correlated with testosterone levels in a Japanese male population. In addition, *Wilmanski et al. (2019)* reported that microbiome diversity is positively associated with androstane-3-17-diol disulfate (a testosterone metabolite). The strongest positive predictor, 5-androstan-3-17 diol disulfate, also correlated with the *Bacteroides* genus, suggesting that the dominance of this taxon was linked to a decline in Shannon diversity. Diet, drugs/pharmaceuticals, geography, birth process, infant feeding methods, stress (exercise, metabolic, and psychological), lifestyle, physical activity, genotype, BMI, and cultural practices determine gut microbiome diversity (*Cresci & Bawden, 2015*; *Koliada et al., 2021*).

Host and gut microbiomes communicate bidirectionally, indicating that they influence each other's functions (*Kim, 2022*). A well-known function of the gut microbiome is to modulate the immune system (particularly the innate immune system) which influences gut-specific diseases and conditions. Obesity-related diseases, liver diseases, inflammatory bowel diseases, and colitis-associated malignancies are associated with the gut microbiome (*Marchesi et al., 2016*). Folates, indoles, secondary bile acids, trimethylamine-$N$-oxide, neurotransmitters (serotonin and gamma-aminobutyric acid), and short-chain fatty acids (SCFA) are vital metabolites in the gut microbiome. Previous study findings show that these microbial metabolites affect the host metabolism by binding to specific nuclear receptors or membranes. SCFA are the most widely investigated metabolites because they are recognized by G-protein-coupled receptors (GPR-41 and GPR-43). Stimulation of these receptors by short-chain fatty acids (SCFA) activates the secretion of peptide-1, such as glucagon or peptide YY, which plays a role in glucose metabolism. The gut microbiome can also influence testosterone levels through the gut-brain axis. Alterations in the composition of the gut microbiome and its active metabolites, such as trimethylamine N-oxide and lipopolysaccharides, cause an imbalance in gastrointestinal homeostasis, resulting in an inflammatory state, glycometabolic disorders, and insulin resistance (*Tanase et al., 2020*). *Lachnospirales* were identified as the cause of substantial increases in plasma glucose and reduced serum insulin in colonized germ-free mice; however, no study has reported the direct effect of this species on the synthesis or metabolism of testosterone in type 2 diabetes

mellitus patients (*Liu et al., 2022*). In addition, gut-derived bacterial endotoxins may trigger testosterone deficiency in men with obesity by impairing testosterone production in the Leydig cells (*Tremellen, 2016*).

Notably, germ-free male mice have a distinct transcriptional profile in the hippocampus compared to conventionally raised mice (*Zhou et al., 2020*). The most interesting difference was in the transcription factor associated with gonadotropin-releasing hormone signaling, suggesting that the gut microbiota may influence androgen synthesis by interacting with the hypothalamic-pituitary-gonadal axis (*Seminara & Crowley, 2008*; *Zhou et al., 2020*). Microbes act on different organs by activating the enteroendocrine cells to produce hormones (*Cani, 2018*). This shows the versatility of the gut microbiome to affect distant organs such as the testicles (*Li et al., 2022*).

The gut microbiome is a key regulator of androgen metabolism, and (*Colldén et al., 2019*; *Yoon & Kim, 2021*) testosterone and dihydrotestosterone can be conjugated by glucuronidation, which increases their water solubility. These glucuronidated androgens are excreted in urine or through bile into the small intestine. An *in vitro* study showed that *Clostridium scindens* has the potential to cleave the side chains of host glucocorticoids, converting them into androgens, and (*Ridlon et al., 2013*) *Colldén et al. (2019)* reported the involvement of the gut microbiome in androgen metabolism and glucuronidation. Gut flora may efficiently deglucuronidate the intestinal glucuronidated testosterone and dihydrotestosterone excreted by the liver (*Colldén et al., 2019*). According to a previous study, germ-free mice have higher glucuronidated testosterone and dihydrotestosterone levels in their intestines than their counterparts (*He et al., 2021*; *Markle et al., 2013*). In contrast, free dihydrotestosterone levels increase in the blood, highlighting the importance of the gut microbiome in preventing excretion and sustaining testosterone levels. Microbiome transplantation reverses this effect (*Aguilera, Gálvez-Ontiveros & Rivas, 2020*; *He et al., 2021*). In addition to their deglucuronidation activity, some specific microbial taxa express steroid-processing enzymes that enable them to directly metabolize steroid hormones (*Li et al., 2022*). Widely distributed environmental bacteria may feed on testosterone and other steroids to eliminate contamination (*Chen et al., 2016*). The endobolome involves the gut microbiota genes, pathways, and enzymes involved in steroid hormone metabolism (*Aguilera, Gálvez-Ontiveros & Rivas, 2020*). The gut microbiome expresses classic steroid-metabolizing enzymes such as 17,20-desmolase, 20a-HSDH, 3a-HSDH, 20b-HSDH, and 5b-reductase (*Aguilera, Gálvez-Ontiveros & Rivas, 2020*; *He et al., 2021*). *Diviccaro et al. (2020)* observed that dihydrotestosterone, 3a-diol, and 17b-E, testosterone metabolites, were substantially higher in the colon than in the plasma. Specific bacteria such as *Butyricicoccus desmolans* and *Clostridiu scindens,* possess steroid-metabolizing enzymes that can convert and use sex steroids (*Ly et al., 2020*; *Maffei et al., 2022*; *Ridlon et al., 2013*).

Another proposed mechanism involving the gut microbiome and androgens is sex hormone-binding globulin (SHBG) metabolism. *Yan et al. (2024)* reported a causal effect of SHBG on the gut microbiota. Higher SHBG levels in men were associated with *Dorea* and *Clostridiales* (*Yan et al., 2024*). In contrast, the gut microbiome may also influence testosterone production and metabolism (*Colldén et al., 2019*). Genome-wide association

studies have shown *Alphaproteobacteria* are associated with high levels of SHBG, a carrier of sex hormones (*Yan et al., 2024*). Microbiome modulation is an emerging therapeutic strategy that affects testosterone levels in men. Altering the composition of the gut microbiome may enhance microbial taxa associated with healthy testosterone levels. Probiotic supplementation, dietary interventions, prebiotic intake, and fecal microbiota transplantation (FMT) are under investigation for their potential to restore or optimize gut microbiota composition and could serve as novel treatment pathways for testosterone-related disorders.

This study has some limitations. First, this review included a small number of studies; only ten studies met the inclusion criteria. While the meta-analysis included a substantial sample size of 35,904 men, the limited number of studies involved may render the conclusions susceptible to modification by future research. Second, the included studies showed significant heterogeneity. The studies included in this review were conducted on six populations: Spanish, Japanese, Ukrainian, Australian, American, Chinese, and Korean. However, these findings may not apply to other ethnicities. Third, we could not perform quantitative analysis. A quantitative analysis with big data from different background, such as different races, would have provided a better and clearer picture of the biodiversity of the microbiomes and the relationship between the gut microbiomes and testosterone levels, limiting our conclusions to being qualitative. Future research should focus on identifying the microbial taxa that are most beneficial for testosterone production and investigating the effects of targeted microbiome interventions in clinical settings. Clinical trials evaluating the impact of specific probiotics, dietary adjustments, and FMT on testosterone levels could provide insights into effective strategies for managing low testosterone levels *via* microbiome modulation. Additionally, exploring personalized microbiome therapies could be a promising avenue for precision medicine, enabling interventions tailored to individual gut microbiome profiles.

## CONCLUSION

In conclusion, most studies reported a significant relationship between the gut microbiome and testosterone levels. Of the ten studies included in this review, a significant relationship was observed between the gut microbiome and testosterone levels. The association between the gut microbiome and testosterone levels appears to be bidirectional, although its extent remains unclear. The gut microbiome has a qualitative relationship with testosterone levels; however, the exact quantitative relationship remains unknown. The findings regarding microbial diversity differed significantly across studies. It would be interesting to determine whether specific types of microorganisms are correlated with higher testosterone levels. This will be useful for designing microbiome transplants for therapeutic purposes. The gut-testicular and gut-brain axes could have significant effects on testosterone production, metabolism, and action. Nevertheless, the relationship between the microbiome and testosterone levels is not straightforward, as comorbidities, obesity, dietary patterns, and lifestyle may influence the gut microbiome. This field is still in its infancy and much work is required to determine the association between the gut microbiome and testosterone

levels. Future research should focus on standardizing microbiome analysis techniques and identifying the microbial taxa most critical for testosterone production.

### List of Abbreviations

| | |
|---|---|
| **BMP** | bone morphogenic protein |
| **CFU** | colony-forming unit |
| **BMI** | body mass index |
| **SCFAs** | short-chain fatty acids |
| **SHBG** | sex hormone-binding globulin |
| **HPA** | hypothalamic-pituitary-gonadal axis |
| **PRISMA** | preferred reporting items for systematic review and meta-analysis |
| **GPR** | G-protein-coupled receptors |

### Funding
The authors received no funding for this work.

### Competing Interests
The authors declare there are no competing interests.

### Author Contributions
- Cennikon Pakpahan conceived and designed the experiments, performed the experiments, analyzed the data, prepared figures and/or tables, authored or reviewed drafts of the article, and approved the final draft.
- Geraldo Laurus conceived and designed the experiments, performed the experiments, analyzed the data, authored or reviewed drafts of the article, and approved the final draft.
- Markus Christian Hartanto conceived and designed the experiments, performed the experiments, analyzed the data, prepared figures and/or tables, and approved the final draft.
- Rajender Singh analyzed the data, authored or reviewed drafts of the article, and approved the final draft.
- Ankur Saharan analyzed the data, authored or reviewed drafts of the article, and approved the final draft.
- Darmadi Darmadi performed the experiments, analyzed the data, authored or reviewed drafts of the article, and approved the final draft.
- Andri Rezano performed the experiments, analyzed the data, authored or reviewed drafts of the article, and approved the final draft.
- Gito Wasian performed the experiments, authored or reviewed drafts of the article, and approved the final draft.

### Ethics
The following information was supplied relating to ethical approvals (i.e., approving body and any reference numbers):

PROSPERO (CRD42022350064)

## Data Availability

This is a systematic review/meta-analysis.

## Supplemental Information

Supplemental information for this article can be found online at http://dx.doi.org/10.7717/peerj.19289#supplemental-information.

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
