# Peer review of "Potential relationship of the gut microbiome with testosterone level in men: a systematic review"

_PeerJ, doi:10.7717/peerj.19289_

## Round 0.1 · original submission · Major Revisions

Based on the reviews received in the time we provided, I recommend revising the submission, considering comments from both reviewers.

Both reviewers' main issues and recommendations are important to improving the submission.

Reviewer 1 ·

Basic reporting

Title & Abstract
Title
In my opinion, the current title effectively captures the main focus of the manuscript, linking the gut microbiome with testosterone levels in men. However, the authors can make more concise. For example, consider: “Gut microbiome and testosterone levels in men: A systematic review.”
Abstract
The abstract provides a solid overview of the aim, methodology, key findings, and significance of the review. However, the authors can improve it by mentioning the specific types of studies reviewed (e.g., observational, cross-sectional, in vitro) and providing a brief note on the methods used to assess the studies, enhancing transparency. For example:
“This systematic review analyzed observational and in vitro studies to evaluate the relationship between the gut microbiome and testosterone levels in men. Quality assessment was conducted using the Newcastle-Ottawa scale, with findings indicating a significant positive correlation...”

Introduction
In my view, the manuscript provides a comprehensive background on the gut microbiome, testosterone production, and the hypothesized relationship between the two, supporting the study’s purpose. However, I recommend adding a brief overview of testosterone’s health implications (e.g., on muscle mass, bone density, and reproductive health) to strengthen the rationale for the study’s importance. To improve clarity, consider also including more on mechanisms that may explain the gut microbiome-testosterone relationship (e.g., microbiome impact on the hypothalamic-pituitary-gonadal axis and androgen metabolism) early in the introduction, which could help readers understand the mechanisms discussed later in the review. Specifically, the authors could:
• Include a short paragraph that highlights testosterone’s role in key areas of male health, such as muscle mass, bone density, reproductive function, and overall well-being. This would establish why testosterone is critical and underscore the significance of investigating factors that might influence its levels.
Suggested addition: "Testosterone is a primary male sex hormone that plays a crucial role in several aspects of health. It is essential for developing muscle mass, maintaining bone density, and supporting reproductive function, including spermatogenesis and libido (https://www.mdpi.com/2218-273X/14/10/1222). Additionally, testosterone influences mood, energy levels, and cognitive function, making it a key factor in overall male health and well-being (https://www.rupahealth.com/post/exercise-and-male-hormones-functional-medicine-insights-for-hormonal-optimization). As testosterone levels decline with age, lifestyle factors and other physiological influences on testosterone production become increasingly significant for maintaining health."
• Expand on potential mechanisms that could explain the link between the gut microbiome and testosterone, such as the gut’s impact on the hypothalamic-pituitary-gonadal (HPG) axis and microbial involvement in androgen metabolism. Adding these details early on can prepare readers to understand the more complex mechanisms discussed later.
Suggested addition: "Emerging evidence suggests that the gut microbiome can influence testosterone production through various mechanisms. One proposed pathway is through modulation of the hypothalamic-pituitary-gonadal (HPG) axis, where gut microbiota may affect the release of gonadotropin-releasing hormone (GnRH) and, consequently, luteinizing hormone (LH), which stimulates testosterone synthesis in the testes (https://www.frontiersin.org/journals/microbiology/articles/10.3389/fmicb.2024.1478082/full). Additionally, certain gut microbes possess steroid-processing enzymes that can directly impact androgen metabolism, contributing to testosterone levels in circulation (Tang et al. 2024; Zou et al. 2024). These potential interactions highlight the complex bidirectional relationship between the gut and testosterone regulation, suggesting a foundational role for the microbiome in male reproductive health."
In my view, the review effectively synthesizes current research on the relationship between the gut microbiome and testosterone in men. I recommend minor revisions to clarify the methodology, add context on broader health implications, and provide clearer future directions to further enhance its impact and rigor.
The manuscript is largely free from grammatical errors. However, minor syntactical improvements in certain sentences could enhance readability. For example:
• The authors should consider rephrasing sentences with multiple clauses to avoid complexity. For instance, rather than "This systematic investigation revealed a significant positive correlation between gut microbiome and testosterone levels in men," they could simplify to, "This investigation found a significant positive correlation between the gut microbiome and testosterone levels in men."
• Some technical terms may benefit from brief clarification or rephrasing to maintain clarity, especially in sections intended for a broader audience. I recommend adding brief definitions for terms like "ceRNA networks" or "microgenderome" where first introduced would be helpful to non-specialist readers.
• The authors should ensure that specific terms, such as "microbiome," "androgens," and "testosterone," are used consistently throughout the manuscript to avoid confusion. In my view, occasionally, terms appear with slight variations that could be standardized for better coherence.
• Commendably, the overall flow of the manuscript is smooth, though transitions between some sections (especially in the results and discussion) could be improved. For instance, I recommend transitioning from the findings on specific microbes to their broader implications on male health to enhance the reader’s understanding.

Figures & Tables
The tables provided (e.g., Table 1 and Table 2) are well-organized and informative, listing key study characteristics and findings clearly. However, the authors should consider adding clearer captions that summarize the main points or correlations illustrated in each table, which would help readers quickly grasp the data’s significance. Additionally, add more descriptive legend to the provided figure, add details on what is being shown

Experimental design

Material and Methods
I found that the methods are well-detailed, following PRISMA guidelines for systematic reviews and clearly outlining the literature search and study selection. Here are some suggestions for improvement. However, here are some considerations for the authors to address:
• In my opinion, the authors should further detail on how quality scores from the Newcastle-Ottawa scale influenced the final inclusion or interpretation of studies would improve replicability.
• It appears that observational studies across various years were included, but no timeframe for when these studies were conducted is specified. The authors can consider commenting on this for transparency.
• A brief mention of whether the sample sizes from the included studies provided adequate statistical power would strengthen the methodology.
• Since a meta-analysis was not feasible, the authors can provide more information on how qualitative data were synthesized. For example, they could explain if the synthesis was based on thematic similarities or microbial taxa reported across studies.

Validity of the findings

Results
In my view, the review adds meaningful value to the field, exploring a relatively novel area of the gut microbiome’s impact on testosterone in men. Commendably, this research advances understanding in a niche that lacks extensive investigation, especially for male-focused microbiome studies.
In my opinion, the review highlights potential correlations and mechanistic hypotheses (e.g., influence via the hypothalamic-pituitary axis and microbial metabolism of testosterone). However, discussing emerging or future approaches, such as the potential for microbiome modulation as a therapeutic intervention for testosterone-related disorders, could enhance its contribution and relevance to clinical practice. The following are my recommendations to the authors:
• Begin by briefly explaining the concept of microbiome modulation and its potential for treating testosterone-related disorders. Mention current or emerging strategies, such as probiotics, prebiotics, dietary interventions, and fecal microbiota transplantation (FMT), that could be investigated to influence testosterone levels by targeting the gut microbiome.
Suggested text: "Microbiome modulation represents an emerging therapeutic strategy that could influence testosterone levels in men. By altering the gut microbiome composition, it may be possible to enhance the microbial taxa associated with healthy testosterone levels. Approaches such as probiotic supplementation, dietary interventions, prebiotic intake, and fecal microbiota transplantation (FMT) are under investigation for their potential to restore or optimize gut microbiota composition and could serve as a novel treatment pathway for testosterone-related disorders."
• Refer to existing studies, if available, that have demonstrated promising outcomes in testosterone levels through microbiome-targeted interventions. For example, mention any research showing that specific probiotics have influenced testosterone levels or studies where gut health improvements led to better androgen profiles.
Suggested text: "Preliminary research has shown that certain probiotics may increase testosterone levels by modulating gut microbiota composition, which, in turn, impacts androgen metabolism pathways. For example, probiotic strains such as Lactobacillus have been associated with beneficial shifts in the gut microbiome, supporting hypothesized improvements in testosterone levels (https://doi.org/10.3389/fmicb.2024.1478082). Similarly, dietary modifications aimed at increasing fiber intake could promote microbiota diversity, potentially favoring bacteria associated with testosterone synthesis."
• Highlight the need for further research to understand which specific microbiota compositions are optimal for testosterone health and how these can be achieved through targeted interventions. Suggest areas for future studies, like identifying specific microbial species that play a significant role in testosterone metabolism, and clinical trials on probiotics or dietary interventions for testosterone-related disorders.
Suggested text: "Future research should focus on identifying microbial taxa most beneficial for testosterone production and investigating the effects of targeted microbiome interventions in clinical settings. Clinical trials evaluating the impact of specific probiotics, dietary adjustments, and FMT on testosterone levels could provide insights into effective strategies for managing low testosterone conditions through microbiome modulation. Additionally, exploring personalized microbiome therapies could be a promising avenue for precision medicine, enabling interventions tailored to individual gut microbiome profiles."
• Conclude with a brief statement on the clinical potential of these therapeutic strategies, emphasizing the significance of microbiome modulation in advancing treatment options for testosterone deficiencies and male reproductive health.
Suggested text: "The potential for microbiome modulation to serve as a therapeutic intervention opens exciting new possibilities for addressing testosterone deficiencies and related health issues in men. As research progresses, microbiome-based treatments may become a viable addition to clinical practice, offering a non-invasive approach to supporting hormonal health and overall well-being."

Discussion
In my view, the findings are presented clearly, with correlations between specific microbes (e.g., Ruminococcus, Dorea) and testosterone well-documented. However, the authors could consider the following in the discussion for additional context:
• Add a paragraph that links testosterone levels, modulated by microbiome health, to broader health outcomes in men. Highlight how fluctuations in testosterone can impact areas such as metabolic health, cardiovascular health, reproductive health, mood, and cognitive function. This would demonstrate the study’s relevance on a global scale, emphasizing the potential for gut microbiome health to play a crucial role in overall male health
Suggested text: "The study’s findings linking specific gut microbiota to testosterone levels have broad health implications. Testosterone is not only critical for reproductive health but also plays a pivotal role in metabolic and cardiovascular health, muscle mass maintenance, and cognitive function (https://doi.org/10.1007/s40618-023-02163-8). Lower testosterone levels, potentially influenced by an imbalanced gut microbiome, have been associated with increased risk of obesity, insulin resistance, and cardiovascular disease, highlighting the importance of microbiome health for global male health outcomes (https://doi.org/10.1093/sxmrev/qeae049). This underscores the potential for gut-targeted interventions to improve testosterone-related health issues, which may reduce the prevalence of associated chronic diseases and enhance overall quality of life in men."
• Elaborate on how testosterone itself can influence microbiome diversity, explaining that the relationship between these variables is complex and bidirectional. Discuss possible mechanisms by which testosterone could alter the gut environment, potentially influencing the prevalence of certain microbial species or overall microbial diversity. Mention studies or theories suggesting that testosterone may affect gut microbiota composition, leading to downstream effects on gut and systemic health.
Suggested text: "While this study emphasizes the impact of gut microbiota on testosterone, evidence suggests a bidirectional relationship where testosterone may also shape gut microbiome diversity. Testosterone can alter the gut environment by modulating immune responses and impacting gut permeability, potentially influencing the abundance of specific microbial taxa (https://doi.org/10.3389/fmicb.2024.1478082; https://doi.org/10.1007/s43032-024-01613-9). For instance, testosterone has been shown to support certain bacterial populations linked to reduced gut inflammation, which could reinforce intestinal health and stabilize microbial diversity (https://doi.org/10.3390/life14060694). This complex interplay indicates that testosterone may not only be a product of gut microbiome interactions but also an active participant in maintaining gut ecosystem balance, suggesting a cyclical relationship between microbiome health and hormonal regulation."

Conclusion
In my view, the conclusions align well with the findings, emphasizing the bidirectional relationship between the gut microbiome and testosterone levels. Commendably, the authors note the need for further research, which is appropriate given the field's emerging nature.
• In my further opinion, to strengthen the conclusions, the authors could:
o Emphasize specific areas where future research could clarify the role of particular microbial taxa in testosterone regulation.
o Recommend methodological improvements for future studies (e.g., standardization of microbiome analysis techniques or the inclusion of diverse populations to address generalizability).

Additional comments

no comment

Reviewer 2 ·

Basic reporting

The manuscript is mostly written in professional and clear English, with adequate technical vocabulary.
Key concepts and terminologies, such as "microgenderome," are explained well.
Some sentences are verbose or poorly structured, making certain sections less concise (e.g., abstract and discussion).
There are occasional grammatical inconsistencies, such as missing articles and awkward phrasing, which could benefit from editorial refinement.
Extensive referencing of past studies related to gut microbiome, testosterone metabolism, and related physiological mechanisms.
Provides a broad context of the gut microbiome’s influence on general health, tying it to the specific focus on testosterone.
Over-reliance on tangential references unrelated to testosterone levels (e.g., studies on skin and neurodevelopment).
A lack of critical evaluation of the cited literature to support the novelty and significance of the study.
The structure follows a standard systematic review format, adhering to PRISMA guidelines.
A PRISMA flowchart for study selection is referenced.
The manuscript addresses the hypothesis of a relationship between gut microbiome and testosterone levels and provides relevant qualitative findings.
The results section aligns with the research question.
The hypothesis could be more narrowly defined, as the manuscript attempts to address several mechanisms and microbial taxa without a clear central focus.
The qualitative analysis lacks depth, as it fails to consistently relate findings back to the hypothesis.

Experimental design

The topic is relevant and aligns with journals focused on the microbiome, endocrine health, and systematic reviews.
The study provides original insights into a relatively underexplored area.
The novelty of findings is somewhat undermined by the lack of quantitative synthesis.
The research addresses a meaningful gap in understanding the bidirectional relationship between gut microbiome and testosterone.
The importance of understanding the "microgenderome" is highlighted.
The research question could be more precisely stated in the introduction.
It is not explicitly stated how this study uniquely fills the knowledge gap beyond summarizing existing literature.
Search strategy, inclusion/exclusion criteria, and data extraction processes are described in detail.
Details regarding the quality and reliability of specific included studies are limited.
Specific search strings or strategies used across databases are not detailed.

Validity of the findings

The manuscript does not assess the novelty of its findings or how they contribute uniquely to the literature.
Future directions for research are proposed, emphasizing the importance of addressing comorbidities and lifestyle factors.
Overgeneralization of the results (e.g., significant impact of gut microbiome on testosterone) is made without robust evidence.

---

## Round 0.2 · Minor Revisions

Based on reviews received, a decision on minor revisions is provided. Please address these for final determination

Reviewer 1 ·

Basic reporting

Title & Abstract
The authors have adequately incorporated the suggestions in the title and made required modifications in the abstract.

Introduction
The authors have made the suggested changes to the manuscript as recommended. A brief overview of testosterone’s health implications, including its effects on muscle mass, bone density, and reproductive health, has been added to strengthen the rationale for the study's importance. Additionally, the manuscript now includes a short paragraph that highlights testosterone’s role in key areas of male health. The addition underscores why testosterone is critical and emphasizes the significance of investigating factors that influence its levels. The suggested text was incorporated to provide readers with a clearer understanding of testosterone’s role in male health and its connection to the study’s focus. Furthermore, the authors have expanded on the mechanisms potentially linking the gut microbiome to testosterone production, including the impact on the hypothalamic-pituitary-gonadal axis and androgen metabolism.

Figures & Tables
The authors have satisfactorily addressed the comments/suggestions.

Experimental design

Material and Methods
The authors have addressed the suggested improvements in the manuscript. They have provided additional details on how quality scores from the Newcastle-Ottawa scale influenced study inclusion and interpretation, enhancing replicability. The authors also clarified the timeframe of the included studies, adding transparency on when they were conducted. A brief mention of sample sizes and statistical power has been added to strengthen the methodology. Additionally, the authors have elaborated on how qualitative data were synthesized, explaining the approach used to analyze thematic similarities and microbial taxa reported across studies.

Validity of the findings

Results
The authors have addressed the recommendations in the manuscript. They have included a brief explanation of microbiome modulation and its potential therapeutic role in testosterone-related disorders, mentioning strategies like probiotics, prebiotics, dietary interventions, and fecal microbiota transplantation (FMT). Additionally, they have referred to existing studies on probiotics influencing testosterone levels and highlighted the need for further research to identify specific microbiota compositions that impact testosterone. The authors also emphasized the clinical potential of microbiome modulation as a novel treatment pathway for testosterone deficiencies and related health issues.

Discussion
The authors have made the suggested changes by adding a brief discussion on the broader health implications of testosterone and microbiome health. They also expanded on the bidirectional relationship between testosterone and gut microbiota.

Conclusion
The conclusions align with the findings, highlighting the bidirectional relationship between the microbiome and testosterone.

Reviewer 2 ·

Basic reporting

I congratulate the authors for their work.
Minor grammatical errors- A final proofreading pass would improve clarity.
Use of technical terms (e.g., "microgenderome") should be defined when first introduced for a broader audience.
Please review references and make sure they are properly cited and updated.
Some details about the relationship between the gut microbiome and testosterone (e.g., microbiome effects on the hypothalamic-pituitary-gonadal axis) should be introduced earlier in the manuscript.
Table legends should be more descriptive.
For PRISMA flowchart there should be more explicit labels describing inclusion/exclusion criteria.

Experimental design

Research question is well-defined, and the authors have explained how the review fills a knowledge gap in understanding the microbiome-testosterone relationship.
The NOS is used for quality assessment, but more details should be provided on how NOS scores influenced study selection or interpretation.

Validity of the findings

The reported studies used different microbiome and testosterone measurement methods, making direct comparisons challenging.
The discussion integrates previous findings, but a clearer direction for future research (e.g., longitudinal studies, microbiome interventions) would enhance impact.

---

## Round 0.3 · accepted · Accept

Congratulations. The revised manuscript was sufficiently revised for the reviewers.